

# Detection and Classification of Laminae in Balloon-borne Ozonesonde Profiles: Application to the Long Term Record from Boulder, Colorado

Kenneth Minschwaner[1], Anthony T. Giljum[2], Gloria L. Manney[3, 1], Irina Petropavlovskikh[4, 5], Bryan J. Johnson[5], Allen F. Jordan[4, 5]

[1]Department of Physics, New Mexico Institute of Mining and Technology. Socorro, New Mexico, USA.
[2]Departments of Applied Physics and Electrical Engineering, Rice University, Houston, Texas, USA.
[3]NorthWest Research Associates, Socorro, New Mexico, USA.
[4]CIRES, University of Colorado, Boulder, Colorado, USA.
[5]Global Monitoring Division, NOAA/ESRL, Boulder, Colorado, USA.

*Correspondence to*: Kenneth Minschwaner (kenneth.minschwaner@nmt.edu)

**Abstract.** We quantify ozone variability in the upper troposphere and lower stratosphere (UTLS) by investigating lamination features in balloon measurements of ozone mixing ratio and potential temperature. Laminae are defined as stratified variations in ozone that meet or exceed a 10% threshold for deviations from a basic state vertical profile of ozone. The basic state profiles are derived for each sounding using smoothing methods applied within a vertical coordinate system relative to the WMO tropopause. We present results of this analysis for the 25-year record of ozonesonde measurements from Boulder, Colorado. The mean number of ozone laminae identified per sounding is about 9±2 (1σ). The root-mean-square relative amplitude is 20%, and laminae with much larger amplitudes (>40%) are seen in ~2% of the profiles. The vertical scale of detected ozone laminae typically ranges between 0.5 and 1.2 km. The lamina occurrence frequency varies significantly with altitude and is largest within ~2 km of the tropopause. Overall, ozone laminae identified in our analysis account for more than one third of the total intraseasonal variability in ozone. A correlation technique between ozone and potential temperature is used to classify the subset of ozone laminae that are associated with gravity wave (GW) phenomena, which accounts for 28% of all laminar ozone features. The remaining 72% of laminae arise from non-gravity wave (NGW) phenomena. There are differences in the both the vertical distribution and seasonality of GW versus NGW ozone laminae that are linked to the contrast in main generating mechanisms for each laminae type.

## 1 Introduction

The day-to-day variability in the vertical distribution of ozone ($O_3$) above a fixed location is often characterized by the presence of transient, stratified features (e.g., Dütsch, 1966). The stratification occurs in the form of layered maxima and minima in the observed vertical profile of ozone, with typical vertical scales between about ~0.2 and ~3 km (e.g., Dobson, 1973; Ehhalt et al., 1983). These features are generally called ozone laminae (e.g., Reid and Vaughan, 1991; Teitelbaum et



al., 1994; Orsolini, 1995; Appenzeller and Holton, 1997; Manney et al., 1998, 2000). Laminar structures in $O_3$ have been observed in both the troposphere and stratosphere, and their generation can be linked to a wide range of mechanisms such as stratosphere-troposphere exchange, tropical and monsoon-related deep convection, gravity waves, differential advection of ozone fields within natural spatial gradients, photochemical production or loss, and advection of urban plumes (e.g.,

Thompson et al., 2011; and references therein). The horizontal scales of laminae can vary significantly, but generally they are observed over 10's to 100's of km, leading to tracer features such as tongues or filaments appearing in quasi-horizontal coordinates (e.g., Randel et al., 1993; Waugh et al., 1996; Bowman et al., 2007; Fairlie et al., 2007; Manney et al., 1998).

The most important dynamical processes that generate ozone laminae in the midlatitude upper troposphere (UT, defined here from ~5 km altitude to the tropopause) are gravity and Rossby waves, convective lofting and detrainment of either high or

low $O_3$ from the lower atmosphere, and intrusions of air masses with high ozone concentrations from the stratosphere (e.g., Langford and Reid, 1998; Thompson et al., 2007b; Selkirk et al., 2010). These generating mechanisms often involve nonlocal dynamics and long-range transport by UT jets, and in some cases the ozone anomalies have been traced back to dynamical events occurring thousands of kilometers from the measurement location (e.g., Vogel et al., 2014, Minschwaner et al., 2015). In the midlatitude lower stratosphere (LS, defined here from the tropopause to ~22 km), gravity waves,

tropospheric intrusions, and differential advection have been identified as drivers of ozone laminae (Teitelbaum et al., 1994; Manney et al., 1998, 2000; Pierce and Grant, 1998; Tomikawa et al., 2002; Pan et al., 2009; Olson et al., 2010). For regions of the atmosphere where the time constants for photochemical production and loss of ozone are longer than dynamical time scales, e.g., at least a week in the UT (Liu et al., 1980) and month in the LS (e.g., Shimizaki, 1984), observations of ozone laminae are evidence of predominantly transport-related phenomena.

A better understanding of the characteristics of ozone laminae and their generating mechanisms is needed in order to fully characterize ozone variability and long-term changes in the upper troposphere/lower stratosphere (UTLS). This understanding is critical to assessing the radiative forcing of climate by ozone and for evaluating the impact of transport on regional air quality. Here, we describe a new method for identifying and classifying ozone laminae from high vertical resolution measurements (~100 m) of ozone, pressure, and temperature. The techniques have been derived and tested on

vertical profiles obtained from balloon soundings, but they can be generalized to other trace gas datasets with sufficient vertical resolution. We present an application of this method to the long-term record (1991-present) of ozonesonde profiles from Boulder, Colorado.

## 2 Dataset

Ozonesonde data are obtained from an in-situ sensor that is flown on a balloon in a package that includes radiosonde and

GPS devices (Komhyr, 1986; Komhyr at al., 1995). An ozonesonde consists of a Teflon air pump and an electrochemical ozone sensor (ECC) with two platinum electrodes in separate cells of potassium iodide solutions with different concentrations. Ambient air is drawn through one cell and the presence of $O_3$ drives chemical reactions that give rise to a



microampere current between the electrodes. A complete description of the ECC ozonesonde is given in Komhyr et al. (1995). Output from the ECC is interfaced to a meteorological radiosonde, which measures air temperature, pressure, relative humidity, and GPS position, and transmits all of the ozone and meteorological data back to a ground receiving station during the ~2 hour balloon ascent. Raw data are taken at ~1 sec resolution during the flight up to the burst altitude, which is

typically at or above 30 km. The precision in ozone mixing ratios in the UTLS region is 3–5 % (1σ) and the absolute accuracy is about 10 %. The combined effect from the sensor time response in the UTLS (~25 s) and the balloon ascent rate (4-5 m s$^{-1}$) gives an effective vertical resolution of about 100 m (Hassler et al., 2014, and references therein). Although some data may be available during the parachute descent phase of the sounding, the ascending flight data are considered the highest quality; mixing ratio profiles used here are from ascent only and are vertically averaged within 100-m thick layers.

Ozonesonde data from Boulder, Colorado (40°N, 105°W, 1.7 km ASL) are available from 1978 to present, with approximately weekly sampling. The data since 1991 have been homogenized by applying instrumental corrections including effects of different buffer solutions (Komhyr et al., 1995; Smit et al., 2007; Deshler et al, 2008) and effects from Teflon air pump efficiencies (Johnson et al, 2002; Deshler et al, 2017). Homogenization of NOAA ozonesonde records to remove instrumental inconsistencies is described by Sterling et al (2018). Prior to 1991, the data are digitized from charts

and are available at 1 minute time resolution (~250 m effective vertical resolution). For consistency in vertical resolution and data quality, we limit our analysis here to the post-1991 ozone data from Boulder.

## 3 Methods

A qualitative description of laminae in a vertical profile of $O_3$ can usually be made by visual inspection, but a quantitative and objective assessment requires a set of criteria for defining ozone perturbations as deviations from some basic state.

Figure 1 shows balloon profiles of ozone mixing ratio ($\chi$) and potential temperature ($\Theta$) observed from Boulder on 10 June 2008. Both profiles contain laminar structures that are easily discernible by eye. For a quantitative analysis of such profiles, we developed an analysis package called RIO SOL (Robust Identification of Observed Signatures in Ozone Laminae), which applies a consistent filtering method (described below) to every profile in order to find basic states for ozone mixing ratio ($\chi_s$) and potential temperature ($\Theta_s$). All perturbations are then defined in terms of differences (i.e., $\chi' = \chi - \chi_s$), and a lamina

is identified when the relative anomaly in ozone is at least 10% (i.e., $|\chi'/\chi_s| \geq 0.1$, see Fig. 1). This amplitude threshold for defining laminae is broadly consistent with previous analyses (Teitelbaum et al., 1994; Grant et al., 1998; Thompson et al., 2007a), but our approach for deriving the basic state is modified to improve detection and identification of laminae located within a few kilometers of the thermal tropopause.

Dobson (1973) and Reid and Vaughan (1991) used a direct sampling method to locate extrema in the vertical profile of

ozone partial pressure. They defined a lamina as a local maximum or minimum in ozone that exceeded 20 nb in peak magnitude, with a full width between 0.2 and 2.0 km (defined with respect to upper and lower "turning points" bracketing the layer). The use of an absolute threshold rather than a relative one to define a lamina was related to their use of ozone



partial pressures, as the change in ozone partial pressure between the troposphere and stratosphere is much less than the corresponding change in ozone mixing ratio. However, a fixed 20 nb threshold in partial pressure is roughly comparable to our 10% threshold in mixing ratio only in the middle stratosphere, between about 20 and 30 km altitude. In the UT, mean ozone partial pressures are much lower (25-40 nb) and the same 20-nb threshold will detect only those anomalies that are

larger than 50-80%. With such a reduced sensitivity, the number of lamina detections in the UT should be smaller than the number of UT laminae we identify using a 10% mixing ratio threshold. Krizan and Lastovicka (2005) and Krizan et al. (2015) used an even larger threshold of 40 nb in peak magnitude to examine "strong" ozone laminae in vertical profiles. Alternatively, Huang et al. (2015) used a Continuous Wavelet Transform (CWT) approach to study ozone laminae in lidar ozone vertical profiles. An interesting feature of the CWT approach is that it does not use a basic state or reference ozone

profile to identify laminae. The laminae amplitude thresholds used by Huang et al (2015) were 10 ppbv in the troposphere and 40 ppbv in the stratosphere, which more closely follows our 10% threshold than the partial pressure criteria discussed above.

Reid and Vaughan (1991) and Huang et al. (2015) compared their methods to "filter and difference" techniques that are broadly similar to the approach used here and in other studies (e.g., Grant et al., 1998; Krizan and Lastovicka, 2005;

Thompson et al., 2007a) and found reasonable agreement in lamina statistics between the two methods. One advantage of the filter and difference approach is that basic states are generated for each profile, as described below, which can provide important information on the contribution of laminae to the overall variability in ozone.

An important drawback to filtering, however, is directly tied to sharp changes in the vertical gradients of ozone and potential temperature near the tropopause. Figure 2 shows observed $\chi$ and $\Theta$ profiles along with two sets of basic state profiles ($\chi_{s1}$,

$\Theta_{s1}$), and ($\chi_{s2}$, $\Theta_{s2}$). The first set of basic states is derived by applying a nonrecursive boxcar filter with a fixed width of 6 km to each observed profile. A fixed-width filter has been employed in a number of previous studies (e.g., Teitelbaum et al., 1994), and for a 6 km boxcar the effective low-pass filter cutoff frequency (~90% level) corresponds to vertical scales of 2-3 km. These basic states are thus smoothed profiles with all features at scales less than ~2.5 km effectively removed (dashed curves in Fig 2). Subsequent differencing with the observed profiles produces anomaly profiles that emphasize laminar

features less than 2.5 km in width. Note that our 0.1-km averaging/sampling grid for the measurement profiles corresponds to an effective Nyquist cutoff for scales less than 0.2 km; thus, this method for laminae identification is focused on features with widths between about 0.2 and 2.5 km. Features that span a vertical range larger than about 3 km occupy a significant fraction of the density scale height, and are more often related to large-scale shifts in air masses than to processes typically associated with the generation of laminae (Reid and Vaughan, 1991). At vertical scales below 0.2 km, however, an

important contribution to ozone variability could result from the presence of very thin laminae. Aircraft measurements of tracer variability have indicated scale-invariant behavior over a wide range of horizontal scales, ranging from 0.2 km to 2,700 km (Tuck et al., 2004). Although this suggests that vertical scales below 0.2 km may be important, the variability in ozone at horizontal and vertical scales less than 0.2 km is not well characterized in the UTLS region.



The method of fixed-width filtering consistently produces an apparent lamina of negative sign near the tropopause level (Fig 2). In this case, the effective $O_3$ and $\Theta$ perturbations are always less than basic state values due to sharp changes in the vertical gradients of both quantities near the tropopause. Rapid gradient changes near the tropopause usually occur on scales less than 2.5 km and therefore are smoothed out in a basic state derived from a 6-km wide filter. A similar effect was noted

by Schmidt et al. (2008) in their analysis of gravity wave activity using GPS temperature profiles. We explored several alternatives for deriving basic states and minimizing tropopause-related artifacts, ranging from piecewise polynomial fitting to the use of climatological means. An important drawback for many approaches is that they cannot accommodate the large degree of variability in the altitude of the tropopause; even seasonal climatologies do not reproduce tropopause variability to the extent needed to remove false laminae detections. We adopted a method for RIO SOL that identifies the primary

tropopause for each profile using the World Meteorological Organization (WMO) lapse rate criterion (e.g., Homeyer et al 2010), and then employs a variable boxcar with a maximum width of 6 km and a minimum width of 1.5 km at the tropopause level. The width varies linearly with altitude within 6 km on either side of the tropopause, such that the boxcar width is symmetric about the tropopause level. As shown in Figure 2, this variable-width filtering method allows the basic states to track sharp gradient changes at the tropopause while still filtering enough small scale variability to identify ozone

laminae in the anomaly profiles. The use of a variable-width-smoothed basic state means that the detection sensitivity for laminae of varying thickness will change with altitude. Away from the tropopause where the filtering width is 6 km, all lamina with vertical scales less than ~2.5 km can be identified. Near the tropopause however, the mean boxcar width is about 2.6 km, which corresponds to a lamina detection threshold width of about 1.5 km. In addition, soundings that contain multiple tropopauses (e.g., Schwartz et al., 2015) are often associated with complex vertical structures in ozone that may not

be fully characterized by this laminae analysis.

The same variable-width smoothing procedure is used to identify laminae in the measured vertical profile of $\Theta$. A lamina detected in potential temperature that is coincident with a lamina in ozone provides evidence that the sampled air parcel was subjected to a vertical displacement associated with gravity wave (GW) activity. This method has been used extensively to examine GW signatures in ozonesonde data (e.g., Teitelbaum et al., 1994; Pierce and Grant, 1998; Thompson et al., 2007a)

and in aircraft measurements of ozone (e.g., Alexander and Pfister, 1995), based on the expectation that

$$\Theta' = \chi' \left[ \frac{\partial \Theta_s}{\partial z} \Big/ \frac{\partial \chi_s}{\partial z} \right] \qquad (1)$$

where $z$ is altitude. Teitelbaum et al. [1996] discuss the general conditions under which this relationship holds, including the condition that the time scales for ozone photochemistry are much longer than the time scales relevant for transport by GW phenomena. They further note that these coincidences are more readily identified by scaling potential temperature perturbations to account for differences in the mean vertical gradients of $\Theta$ and $\chi$,

$$R(z) = \left( \frac{1}{\chi_s} \frac{d\chi_s}{dz} \right) \Big/ \left( \frac{1}{\Theta_s} \frac{d\Theta_s}{dz} \right) \qquad (2)$$




A similar scaling was employed by Ehhalt et al. (1983) to compare equivalent vertical displacements obtained from measured variances in long-lived stratospheric gases.

One of the most straightforward approaches for identifying coincidences in ozone and potential temperature laminae involves the spatial correlation between vertical profiles of $\chi'/\chi_s$ and $R\Theta'/\Theta_s$ over a limited vertical domain, for example within 5-km wide sampling windows [e.g., Teitelbaum et al., 1994]. Figure 3 shows a set of relative anomaly profiles along with the magnitude of the correlation coefficient computed within a 5-km wide vertical window centered at the given altitude. A correlation threshold of $r > 0.7$ has been shown to be a reliable indicator for GW induced laminae in ozone (e.g., Pierce and Grant, 1998). One complication with this approach arises when multiple laminae appear within the same correlation window. For example, in Figure 3 the central altitudes of laminae labeled 2 through 5 are close enough to cause interference in gauging the true correlation within individual laminae. We adopted a different approach for RIO SOL, by correlating over the extent of the laminar feature (where the amplitude exceeds 10%) or over a 2-km window, whichever is larger. Sensitivity experiments indicate that our approach for deriving basic states and for correlating over more limited vertical domains is more consistent with previous analyses if we adopt a threshold of $r \geq 0.65$. In Figure 3, RIO SOL detects GW ozone lamina (labeled 1 and 3) near 9.5 and 14 km altitude with the application of the $r \geq 0.65$ threshold. Laminae for which ozone anomalies are not significantly correlated with scaled potential temperature anomalies ($r < 0.65$) are classified as non-gravity wave (NGW) laminae, as there is no evidence that the generation mechanism is associated with GW activity.

In order to examine the detection sensitivity for laminae, we constructed a set of 150 simulated ozone and temperature profiles using climatological values representing tropical, midlatitude summer, and midlatitude winter means (Anderson et al., 1986), and introduced localized ozone perturbations at random altitudes and over a random sample of amplitudes and widths. The perturbations were either triangular or Gaussian in shape. The simulated profiles were then analyzed using RIO SOL. Figure 4 shows two examples from the simulations and analysis. In the first example, three laminae were introduced to a tropical basic state and RIO SOL accurately characterized the anomalies (Fig 4, top). The amplitudes, widths, and central altitudes of positive and negative laminae are derived to within a few percent.

The second example is taken from a midlatitude simulation and highlights one of the weaknesses of the filter and difference approach. A large amplitude lamina (e.g., near 18.5 km in Fig 4, bottom) can shift the derived basic state enough to produce pairs or triplets of laminae in the derived perturbation profile. This effect can be seen in Fig 4 by the appearance of a false positive lamina near 17 km, which is an artifact produced by the large negative laminae immediately above it. False lamina detections accounted for nearly 25% of the total number of laminae identified in the simulations, with no altitude dependence in the number of false detections. This is in contrast to the fixed 6-km width smoothing method described in Section 3, which generates false laminae detections at the tropopause in nearly every profile. As expected, the proportion of false detections grows to nearly 50% as the lamina amplitude threshold is reduced from 10% to 5%. It should be noted that the overall false detection rate is a direct consequence of how these simulations are designed. Most false detections are associated with a single large amplitude lamina in the simulation. Although we expect that the true number of false





detections in observed ozone profiles is likely smaller, this effect is impossible to quantify because there is no way to directly observe the basic state.

In terms of positive identification of true features, the detection rate was 79% for all simulated laminae with amplitudes larger than 10% and widths between 0.2 and 2.5 km. The detected fraction was degraded to about 60% for those laminae within ±2 km of the tropopause, primarily because of the reduced filtering width used to derive the basic states near the tropopause. At all other altitudes, roughly half of the non-detections arose from simulations involving two or more laminae occurring in close proximity (within a few kilometers altitude) which were counted as a single lamina in the identification process. Most of the remaining non-detections were due to simulated laminae with amplitudes just above the 10% threshold that were not counted because the derived amplitudes for these laminae fell just below 10%. On average, there is a 2-4% low bias in derived amplitudes using our version of the filter and difference method, and there is a tendency to underestimate widths by 0.1 to 0.2 km compared to the simulated inputs. Both of these small biases can be seen in some of the simulated laminae shown in Figure 4, and they are an inevitable result of low-pass filtering to determine the basic state. Laminae altitudes are, however, accurately identified to within ±0.1 km.

## 4 Results

### 4.1 Overall Statistics for Ozone Laminae

The RIO SOL analysis was applied to 1138 ozone soundings from Boulder, Colorado. As discussed in section 2, these were obtained on a ~weekly basis between the years 1991 to 2015. A total of 9952 ozone laminae were identified, corresponding to a mean number of 8.7 laminae per sounding. The variability in the number of lamina per sounding was very close to a normal distribution about the mean, with a standard deviation of 2.3 laminae. There were no soundings with fewer than 2 or with more than 16 laminae detections.

There are considerable differences in the frequency of lamina detections with respect to altitude, season, and lamina type. The number of laminae observations per sounding within 1-km thick altitude bins relative to the WMO tropopause is shown in Figure 5. The occurrence frequency for all ozone laminae maximizes near the tropopause and is roughly evenly distributed above and below the tropopause. Over 60% of all laminae were observed within 5 km of the tropopause. Figure 5 also displays occurrence frequencies for GW and NGW laminae, segregated by positive (+GW, +NGW) or negative anomalies (-GW, -NGW) with respect to the basic states. The most common lamina type is -NGW, which accounts for nearly half of all laminae detected outside of the tropopause region. Within 2 km of the tropopause, higher frequencies of +GW and –GW lamina contribute a more significant amount to the total. Over all altitudes, 28% of all laminae are the GW type and 72% are NGW laminae. Negative anomaly laminae outnumber positive anomaly laminae at most levels, and overall we detect about 15% more negative anomaly laminae.

Two of the most important characteristics of a laminar structure are its amplitude and thickness (or width). Figure 5 includes panels for the vertical distributions of the root mean square (RMS) amplitudes and widths of detected lamina. For both RMS



amplitudes and widths, no significant differences were found between positive and negative anomaly laminae. The amplitude of a lamina is defined by the mean of the perturbation taken over the full altitude range where the perturbation magnitude exceeds the 10% minimum threshold. Figure 5 shows that RMS amplitudes are closely matched between GW and NGW laminae, with values between 15% and 20% in the troposphere and an overall tendency for larger amplitudes in

the LS. The mean RMS amplitude taken over all altitudes and laminae type is 20%. The amplitude distribution is skewed by the presence of larger amplitude (>40%) laminae that are seen in ~2% of the soundings. These large-amplitude laminae are most often observed in the LS.

We define laminae widths by the continuous range of altitude levels where the 10% minimum amplitude threshold is met in the anomaly profile. Figure 5 shows that average widths for laminae at Boulder are about 1 km in the troposphere,

decreasing to ~0.7 km near the tropopause and increasing again in the stratosphere (as noted below, a large fraction of this variation is a result of the detection method). The largest mean widths (~1.4 km) were found for NGW laminae occurring about 5 km above the tropopause. As with amplitudes, no significant differences were found between the mean widths of positive and negative anomalies.

The frequency distribution of laminae widths is shown in Figure 6. It varies with relative altitude as expected because of the

variation in basic state smoothing parameters with respect to the tropopause. At relative altitudes larger than ±5 km, the distribution has a significant tail where lamina widths up to 2.5 km are observed. Closer to the tropopause, we find a truncation of the width distribution around 1.5 km results from changing the filtering parameters for the basic states. If we assume that distribution of laminae widths far from the tropopause is representative of the entire profile, then we can estimate that roughly 16% of ozone laminae near the tropopause may not be identified because their widths are larger than

our upper detection limit in this region. This undetected fraction estimated from Figure 6 is consistent with the laminae simulations discussed above, in which the fraction of undetected laminae increased by 19% within 2 km of the tropopause. It should be noted, however, that the simulated negative lamina at 18 km in Figure 4 is within 1.2 km of the tropical tropopause, and it is accurately characterized by RIO SOL. On the narrow side of the width distribution, extrapolation of the smoothly decreasing widths below modal values of ~ 0.4 km at all altitudes yields a detection loss rate of about 2-4% due to

lamina with widths narrower than 0.2 km.

Figure 6 also shows the frequency distribution of laminae amplitudes. Note that the distribution is truncated at 10% by the minimum threshold used for lamina detection. Sensitivity runs using smaller minimum thresholds indicate a significant number of small (5-10% amplitude) laminae fall below our 10% minimum, as indicated in Figure 6. A mean of about 12 laminae are detected per sounding using a 5% minimum amplitude threshold, an increase of about 40% laminae over using a

10% threshold. As discussed above, the fraction of false detections grows with decreasing amplitude threshold, so that the frequency estimate for amplitudes between 5% and 10% shown in Figure 6 should be regarded as an upper limit on the true occurrence of small-amplitude laminae.



## 4.2 Basic States and Ozone Variance

As discussed in Section 3, the filter and difference approach produces basic state profiles that can be used to quantify the fraction of overall variability in ozone attributable to laminar features in the profile. Figure 7 shows the climatological means and standard deviations of basic states for each of the four seasons (December-January February=DJF, etc.). These are displayed in altitude coordinates rather than tropopause-relative coordinates in order to highlight seasonal differences. Because of the filtering methods used to derive these basic states, all of the basic state variability in the UT arises from ozone changes occurring on vertical scales larger than 2-3 km. The mean basic states are nearly identical to climatological seasonal means obtained from the raw data, so that many of the expected seasonal effects are seen in the mean basic state profiles. For example, larger ozone mixing ratios occur during winter/spring in the 10 to 20 km altitude range as a result of stratospheric transport and seasonal changes in the tropopause height. This seasonality can be quantified by the standard deviation of the seasonal basic states at each altitude, as shown in Figure 7, and can be directly compared with the intra-seasonal (i.e., within each season) variability derived from the standard deviation of each of the seasonal mean profiles. As noted above, the seasonal component of ozone variability is largest between 12 and 16 km altitude. However, the intra-seasonal basic state variability tends to follow the climatological tropopause and maximizes in the UT about 1-2 km below the WMO tropopause. During winter and spring, there is a secondary increase in LS variability (12-15 km), which is likely to be related to deep stratospheric intrusions of tropical/subtropical air like those investigated by Reid et al. (2000).

In summary, we expect contributions to the total intra-seasonal ozone variance arising from three types of features: (i) detected laminae with widths between 0.2 and 2.5 km and amplitudes greater than 10%, (ii) all variations with larger vertical scales (> 2.5 km), and (iii) small amplitude (< 10%) features across all vertical scales. Assuming that the total intra-seasonal ozone variance $\sigma_T^2$ can be effectively decomposed into these components, then

$$(\sigma_T/\chi)^2 = (\bar{A})^2 + (\sigma_S/\chi)^2 + (\bar{\delta})^2 \qquad (3)$$

where $\bar{A}$ is the RMS amplitude of detected laminae, $\sigma_S^2$ is the variance in the basic state profile, and $\bar{\delta}$ is the normalized variance due to small (<10%) amplitude features. The left side and the two largest terms on the right side of eq. (3) are shown in Figure 8 for the seasons of DJF and JJA. Raw ozone data was used to calculate $\sigma_T^2$, while $(\bar{A})^2$ and $\sigma_S^2$ were derived from the laminae amplitudes and basic states outputs of the RIO SOL analysis. Figure 8 shows that the total variance in ozone is generally controlled by large vertical-scale changes near and immediately below the tropopause. Laminae make up an important fraction of the total variance, however, and these features can be the dominant mode of ozone variability in the middle troposphere and lower stratospheric regions. The contribution from small amplitude features $(\bar{\delta})^2$ was calculated from the results of the threshold sensitivity experiments discussed above (or could also be estimated as a residual from eq. (3)), and this contribution is typically between 0% and 5% of the total variance. Figure 8 also shows seasonal/altitude means of the contributions from all three terms on the right side of eq. (3), indicated within boundaries of a coordinate system defined by amplitude and vertical scale of ozone variations. Our results indicate that, on average, more



than half of the intra-seasonal variance in ozone between 5 and 25 km altitude is due to large-scale changes in the basic state, and that slightly over one-third of ozone variations are due to laminar features that are identified by RIO SOL.

## 4.3 Boulder Laminae Climatology

We examine here how the frequency of detected laminae varies with altitude and season. Figure 9 shows monthly mean
frequencies relative to the WMO tropopause for GW and for NGW ozone laminae. Consistent with Figure 5, GW laminae are most often observed within 2 km of the tropopause, whereas NGW laminae are more evenly distributed throughout the UTLS. Throughout most of the year, the GW frequency distribution maximizes slightly above the tropopause while the NGW frequency is larger below the tropopause. This situation reverses during the months of June-September, when more GW laminae are detected below the tropopause and most NGW laminae are found above the tropopause.
Climatologies of gravity wave momentum fluxes indicate a wintertime maximum in the LS for this location (e.g. Geller et al., 2013). However, it should be noted that while our ozone GW laminae climatology is expected to be related to the overall level of gravity wave activity, there are other factors that can influence whether a GW lamina in ozone is generated at the location of an ozonesonde profile in the first place, and then whether it will be detected by RIO SOL. For example, our simulations suggest that GW laminae are more readily detected and identified as such in regions where background vertical
gradients of ozone and potential temperature are both large, such as in the LS region.

The distribution of NGW laminae stands in stark contrast to that of GW laminae, and, as noted in the introduction, the generating mechanisms for NGW laminae are much more uncertain. Previous analyses have used maximum correlation threshold criteria between ozone and potential temperature to infer the influence of Rossby waves on ozone (e.g. Pierce and Grant 1998; Thompson et al 2007a). While we have not adopted this approach in RIO SOL, it is plausible that a significant
fraction of NGW laminae are generated in processes associated with Rossby wave activity. The seasonality of Rossby wave breaking in the UTLS at northern midlatitudes (e.g., Hitchman and Huesmann, 2007) is similar to the seasonality of UT NGW ozone laminae seen in Figure 9. Rossby wave breaking and stratosphere/troposphere exchange (STE) are both prevalent along the flanks of UT jets (Gettelman et al., 2011 and references therein), and studies of jets using meteorological reanalysis data show a maximum in subtropical UT jet frequencies near 30$^{\circ}$ latitude from November through April in the
Northern Hemisphere (Manney et al., 2011; 2014). An associated phenomena of multiple tropopause events, which are often linked to tropopause folds and extratropical STE (e.g., Sprenger et al., 2003), also show maximum frequencies during December-March on the northern flank of the subtropical jet maximum (e.g., Manney et al., 2014), and these events are responsible for a significant fraction of the variability in ozone and other trace gases in the tropopause region (Schwartz et al., 2015).
Mechanisms leading to the maximum in NGW laminae frequency in the LS during summer are more uncertain. Impacts on the composition of the midlatitude summer LS have been demonstrated from monsoon-related dynamics (e.g., Randel et al., 2010), deep summertime convection (e.g., Weinstock et al., 2007), and meridional transport from the tropical UT (e.g., Bönisch et al., 2009). Clearly, more work is needed to understand the full range of dynamical and chemical processes that





may produce the range of laminar ozone features seen in these balloon profiles and in other ozone measurements. The development of methods to further classify NGW laminae and to associate mechanisms with their generation is a focus of ongoing work using RIO SOL.

## 5 Conclusions

We have described the RIO SOL analysis package for characterizing ozone laminae in balloon soundings and presented an analysis of the ~25 year ozonesonde dataset from Boulder, Colorado. RIO SOL involves an adaptation of the filter and difference approach used in previous studies of ozonesonde profiles, in which a unique basic state is generated for each ozone profile and laminae are identified as deviations from this basic state. The major improvements in RIO SOL include methods for improved sensitivity in identifying GW laminae using potential temperature from each sounding, and for

avoiding false detections of GW laminae near the thermal tropopause. The vertical gridding of the ozonesonde data and the filtering method constrain the range of vertical scales for identified laminae to between 0.2 and 2.5 km. Simulations indicate that RIO SOL can reliably identify most of the ozone laminae with relative amplitudes greater than 10%, and virtually all laminae above 20% amplitude.

The mean number of ozone laminae observed per sounding at Boulder is about nine. This is much higher than the number

(1-2) reported by Reid and Vaughan (1991) and Krizan et al (2016) from analyses of northern midlatitude soundings, but as noted in the introduction, the fixed detection threshold used in these studies emphasized stratospheric ozone laminae at the expense of tropospheric laminae. Huang et al. (2015) used separate thresholds for the troposphere and the stratosphere and found a mean of 2.5 laminae per profile in the ozonesonde dataset from Huntsville, AL. This is still significantly less than the number of laminae detected at Boulder. The method used by Huang et al. limited the smallest scale of detected laminae

to 0.5 km as compared to the 0.2 km limit used here; thus, it is likely that the difference can largely be explained by the detection of laminae at smaller scales and with smaller amplitudes (due to the use of a relative amplitude threshold in RIO SOL). The root-mean-square laminae amplitude at Boulder is about 20% relative to the basic state. This mean amplitude does not vary significantly with altitude. It is slightly larger than the ~15% mean amplitude determined by Pierce and Grant (1998) for ozonesondes from Wallops Island, VA, although their identification method could have been impacted by

interference near the tropopause. The mean width of ozone laminae, defined by the altitude range in which the anomaly amplitude exceeds 10%, varies between 0.7 and 1.4 km depending on the altitude relative to the tropopause and is generally smallest near the tropopause. Some of the variation in mean width with altitude is a result of the filtering procedure used in RIO SOL.

Laminae statistics have been examined on a tropopause-relative altitude grid rather than on standard altitudes relative to the

surface. This gridding better delineates the subset of laminae generated by primarily tropospheric mechanisms from those resulting from mainly stratospheric phenomena, and facilitates the identification of STE processes. The occurrence frequency for ozone laminae shows a distinct maximum within ~2 km of the tropopause and is nearly symmetric above and



below the tropopause. GW laminae make up about one-third of all ozone laminae. These are most often detected near the tropopause in the lower stratosphere. NGW laminae are more abundant, and negative NGW are the most dominant laminar feature throughout the UTLS region.

The total variance in the Boulder ozonesonde dataset was decomposed into terms representing changes in the ozone basic state, changes due to the presence of ozone laminae, and changes due to weaker (<10% amplitude) features. Large scale changes in the basic state account for 60% of the total intra-seasonal ozone variance. The magnitudes of intra-seasonal variations in basic states are comparable to those of the seasonal cycle. Laminae detected by RIO SOL are responsible for 37% of the total variance in ozone, and the remaining 3% is estimated to originate from smaller scale features. Although they are not the dominant form of ozone variability, laminae must be considered in order to quantify ozone variability and trends in the UTLS region. Future research in this area should be directed towards methods for obtaining global laminae datasets, either from satellite measurements or from ozonesonde networks, and for developing improved techniques for unambiguous identification of mechanisms responsible for generating NGW ozone laminae.

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



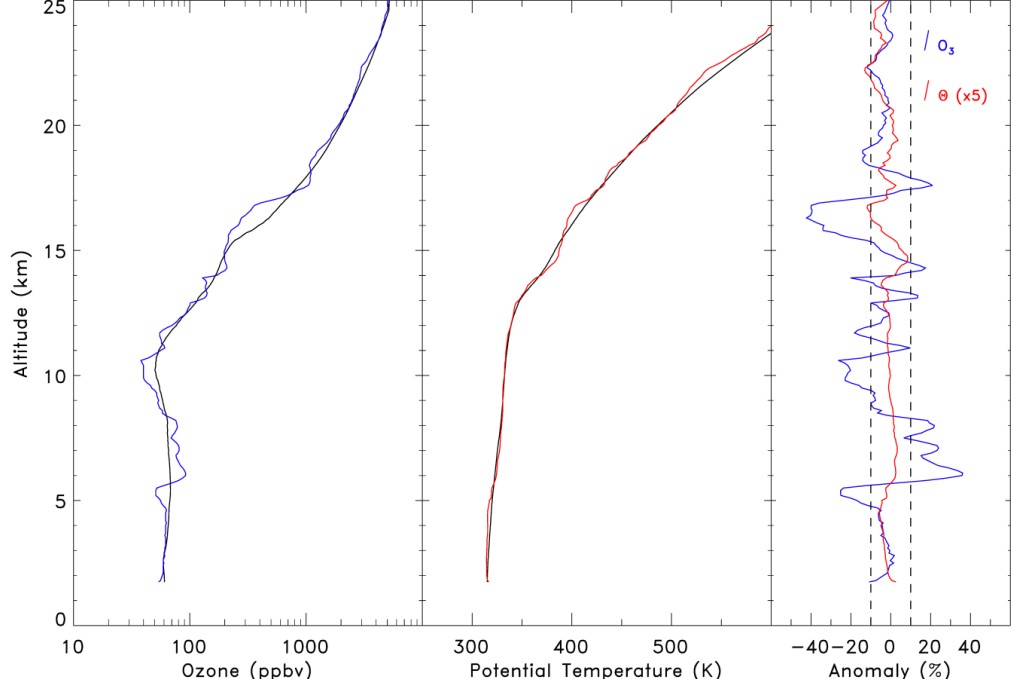

**Figure 1: Vertical profiles of ozone (left, solid blue) and potential temperature (middle, solid red) measured from Boulder, CO on 10 June 2008. The respective basic states are indicated by solid black curves in both panels. The far fight panel shows relative anomalies based on differences between the measured and basic state profiles for ozone (solid blue) and potential temperature (solid red, scaled by a factor of 5). Dashed vertical lines denote the +-10% threshold in ozone anomaly that is used to identify lamina.**

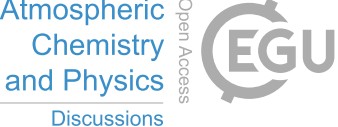



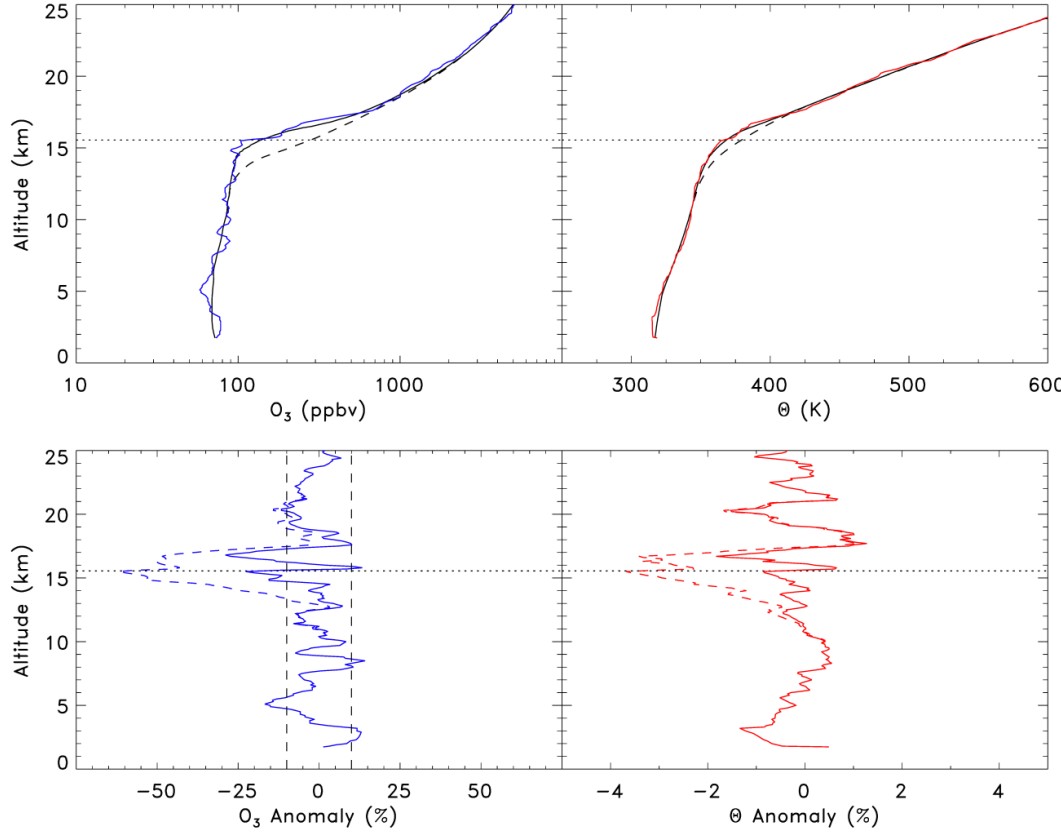

**Figure 2: Vertical profiles of ozone (top left, solid blue) and potential temperature (top right, solid red) measured from Boulder, CO on 6 August 2008. Both panels also show basic states calculated using a fixed-width filter (dashed), and a variable-width filter (solid), as described in the text. The bottom two panels show anomalies based on the fixed-width (dashed) and variable-width (solid) filters for ozone (bottom left, blue), and potential temperature (bottom right, red). Vertical dashed lines for the ozone anomaly indicate the +-10% threshold for laminae detection, and the horizontal dotted lines in all panels indicate the height of the lapse rate tropopause determined from this sounding.**



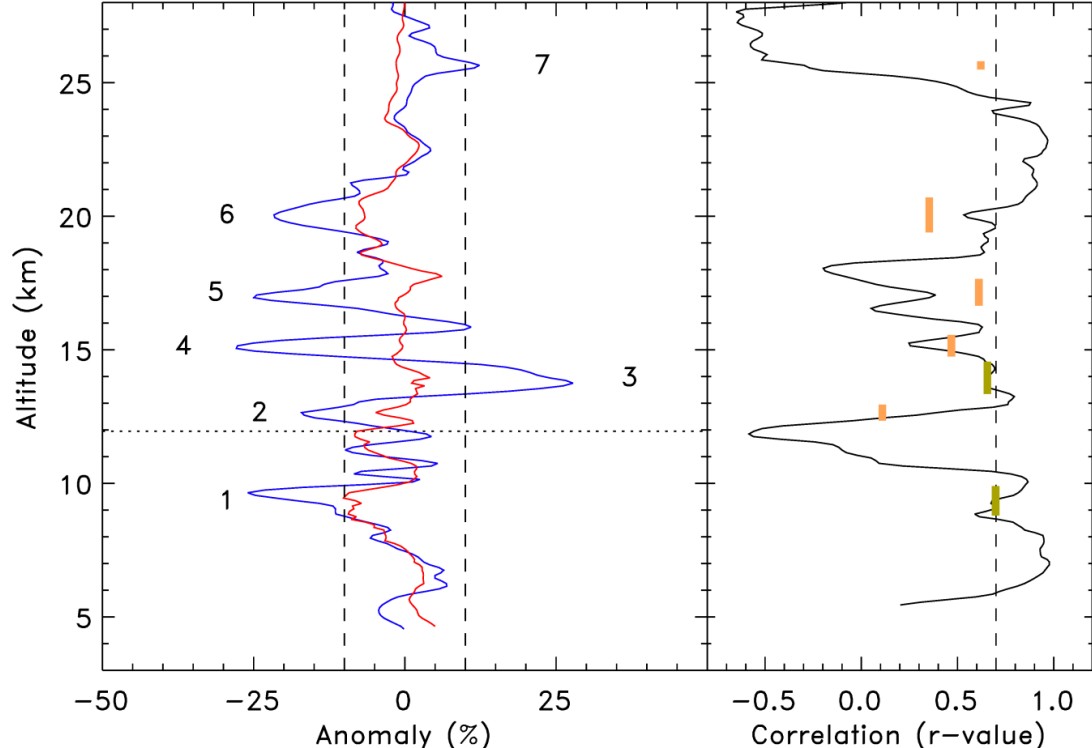

**Figure 3: Left panel shows vertical profiles of ozone (blue) and scaled potential temperature anomalies (red) from a Boulder ozonesonde sounding on 4 May 2006. Vertical dashed lines represent a 10% amplitude threshold for defining laminae, and seven identified laminae are indicated by number. Right panel shows the correlation coefficient calculated between ozone and potential temperature anomalies using a 5-km wide sliding vertical window (solid curve), and alternatively from windows centered on individual laminae (orange and green bars). The dashed vertical line indicates the 0.7 threshold used for identifying gravity wave ozone laminae with the 5-km sliding window technique. For correlations over individual laminae, a 0.65 threshold value is adopted, and laminae meeting or exceeding this threshold are indicated by the green bars and classified as GW laminae, while correlations below 0.65 are indicated by orange bars and classified as NGW laminae.**



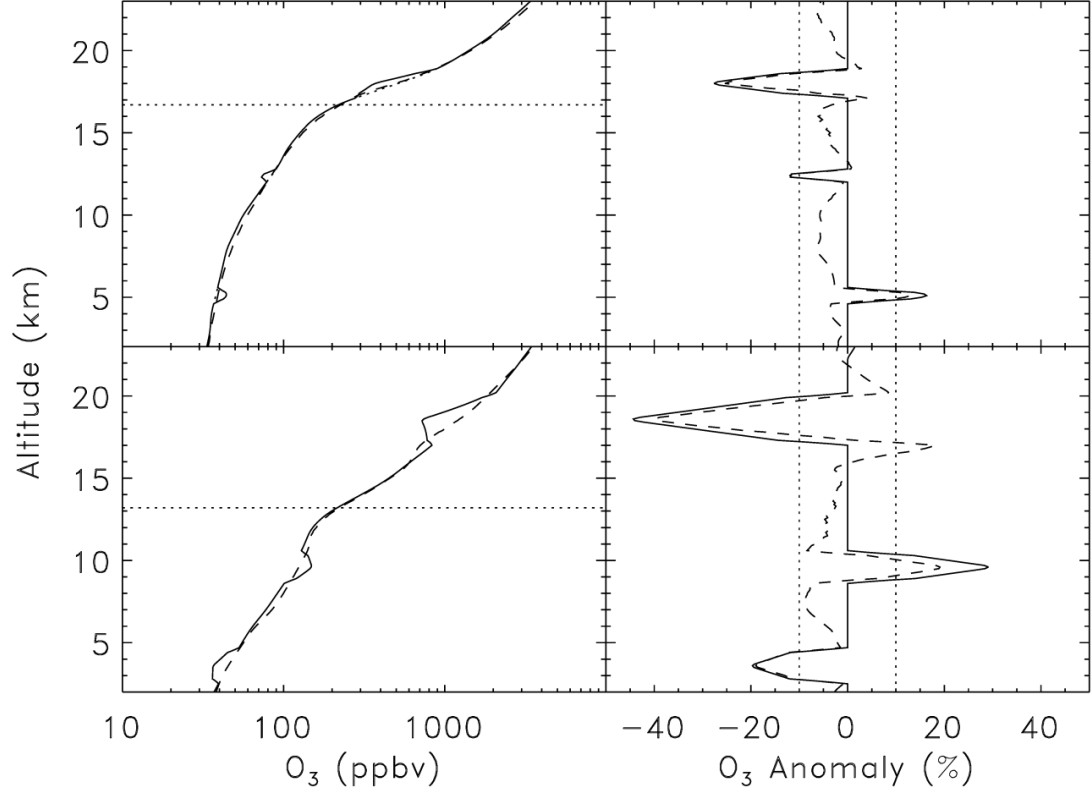

**Figure 4: Two simulated ozone profiles with randomly placed laminae applied to a tropical basic state (top panels), and to a midlatitude basic state (bottom panels). In each case, the left panel shows the simulated ozone (solid) and the derived basic state (dashed), while the right panel shows the actual anomaly profile for each simulation (solid), along with the derived anomaly profile (dashed). Horizontal dotted lines in the left panels denote the tropopause level, and vertical dotted lines in the right panels denote the 10% anomaly threshold used for detecting ozone laminae.**



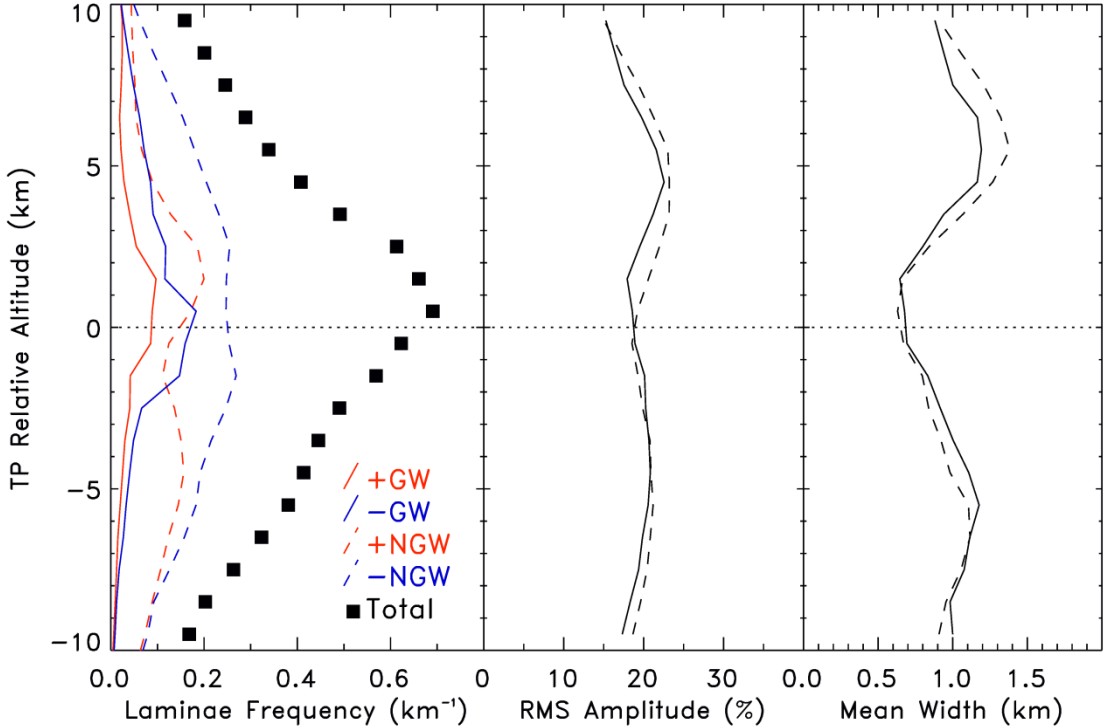

**Figure 5: Vertical profiles of ozone laminae characteristics in altitude coordinates relative to the WMO tropopause. Left panel shows laminae frequency as the number of laminae detected per sounding within 1-km wide altitude bins. Black squares are for all laminae types and signs. Solid lines show frequencies of GW laminae, dashed lines indicate NGW laminae, and red and blue colors indicate positive and negative anomalies, respectively. The middle and right panels show profiles of RMS amplitudes and mean widths, respectively. RMS amplitudes are derived from the square of the mean relative anomaly within each lamina, and averaged over all laminae detected within corresponding relative altitude bins. Widths are defined as the full altitude range where the anomaly amplitude exceeds 10% (along consecutive 100-m sampling intervals), with a minimum restriction of 0.2 km. As with the left panel, solid and dashed curves denote GW and NGW laminae, respectively.**





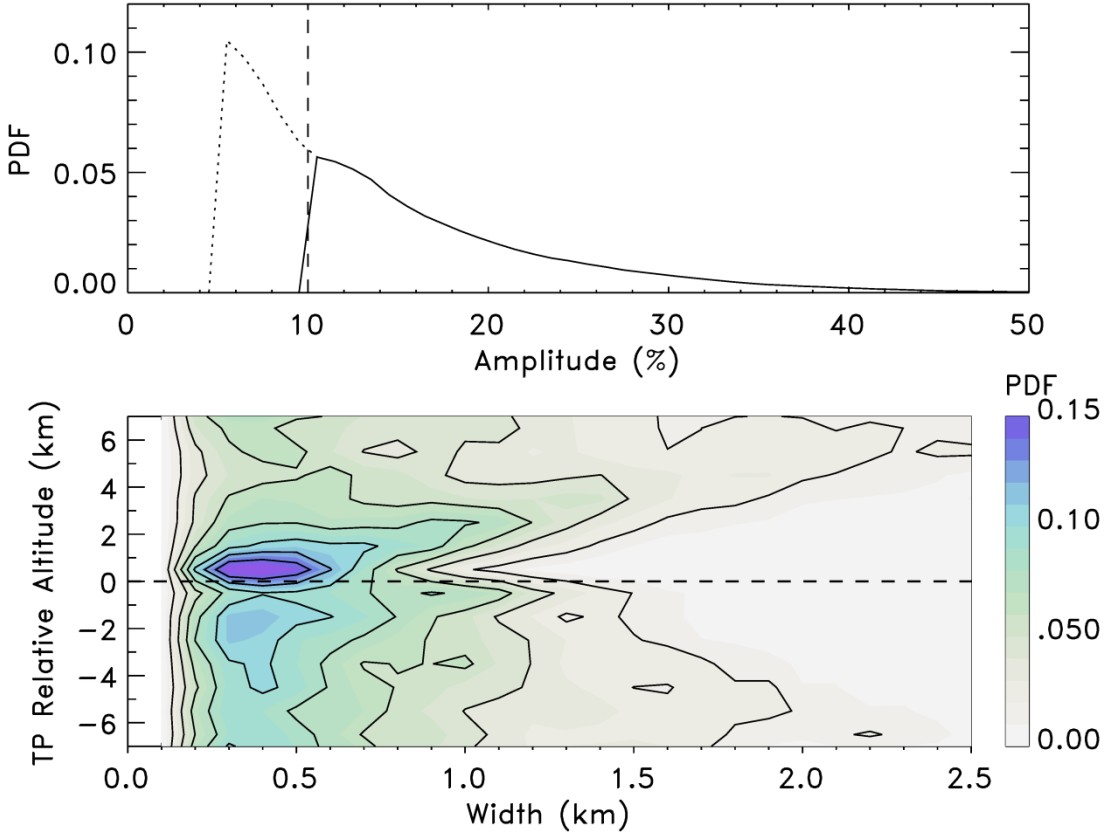

**Figure 6:** Probability density function (pdf) of the amplitude distribution for all ozone laminae (top), and contour pdf of width distribution for ozone laminae versus altitude relative to the WMO tropopause (bottom). For the amplitudes, the standard 10% detection threshold is indicated by the vertical dashed line and the solid curve shows the amplitude distribution using this criterion. The dotted curve shows the extension of the amplitude pdf if a 5% detection threshold is employed. For the width pdf, contours are separated by 0.02, and the black dashed line indicates the tropopause level.



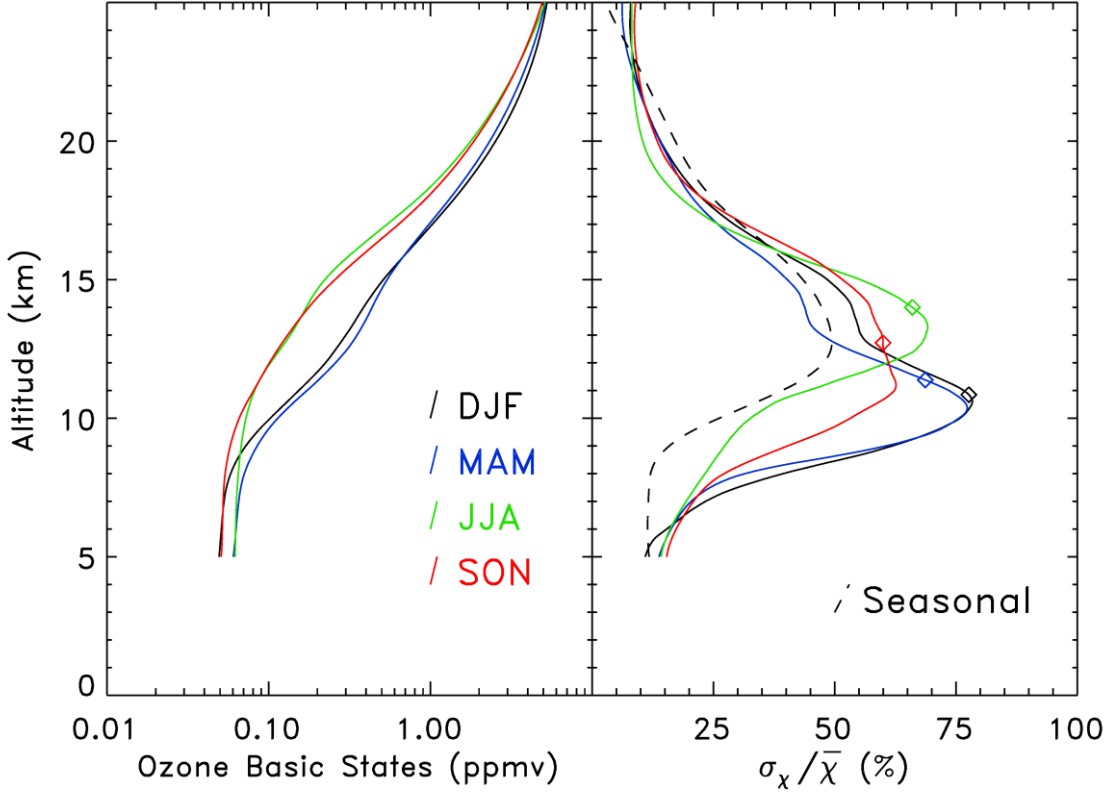

**Figure 7: Seasonal means of ozone basic states profiles from Boulder over the years 1991-2015 (left), and normalized standard deviations of the basic states (right). The means and intra-seasonal standard deviations were taken over the periods December-February (DJF, black), March-May (MAM, blue), June-August (JJA, green), and September-November (SON, red). The standard deviations were normalized by mean values at each altitude, and seasonal mean tropopause heights are indicated by colored triangles. Also shown is comparable magnitude of the seasonal variation in basic states (dashed curve, right panel), estimated from the variance in the four seasonal basic states and normalized by the annual mean. Note that this seasonal magnitude is equivalent to $A_p/\sqrt{2}$ for a cosine seasonal variation, where $A_p$ is the peak relative amplitude.**




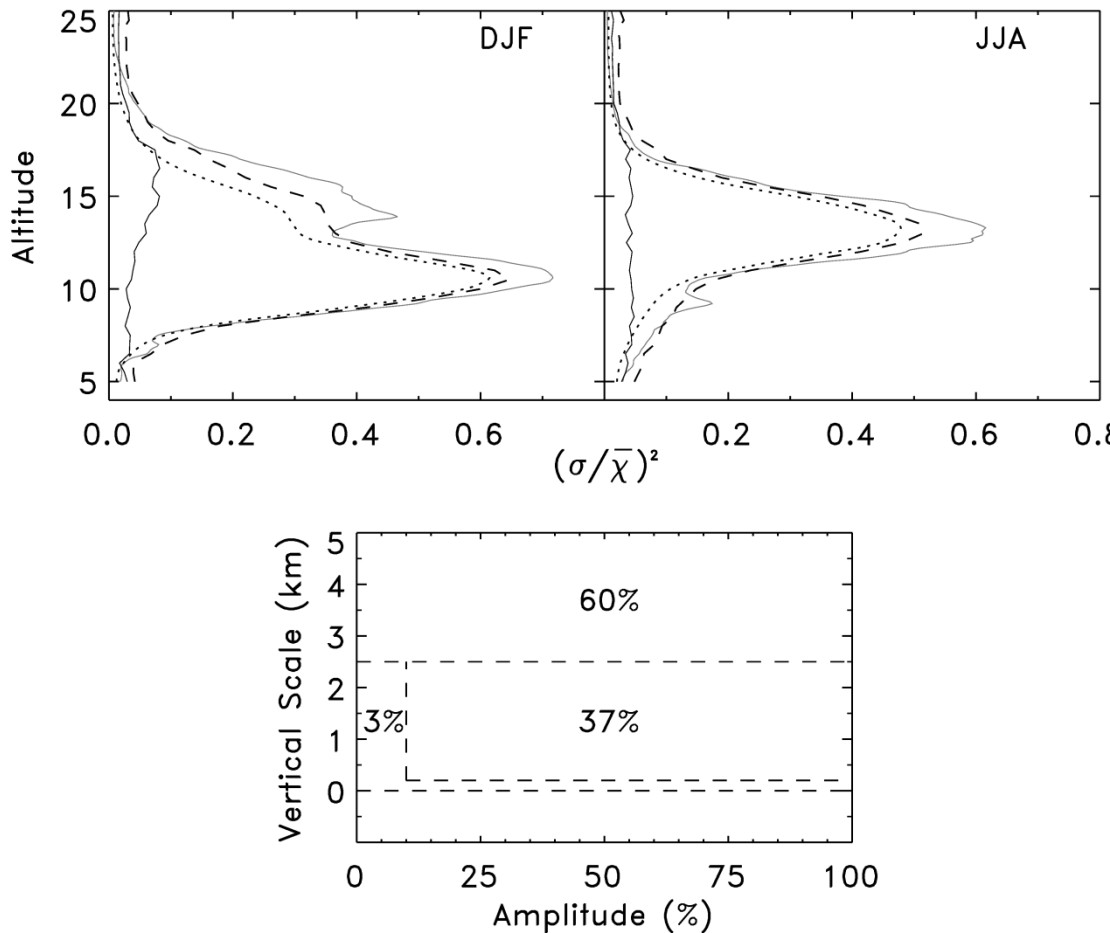

**Figure 8: Normalized seasonal variances in ozone for DJF (top left) and JJA (top right). Plotted are the normalized total variance (solid gray), basic state variance (dotted), laminae variance (solid black), and the sum of basic state and laminae variances (dashed). Bottom diagram shows the annual mean contributions to the total variance between 5 and 25 km altitude, as a function of amplitude and vertical scale of features in the ozonesonde profile.**





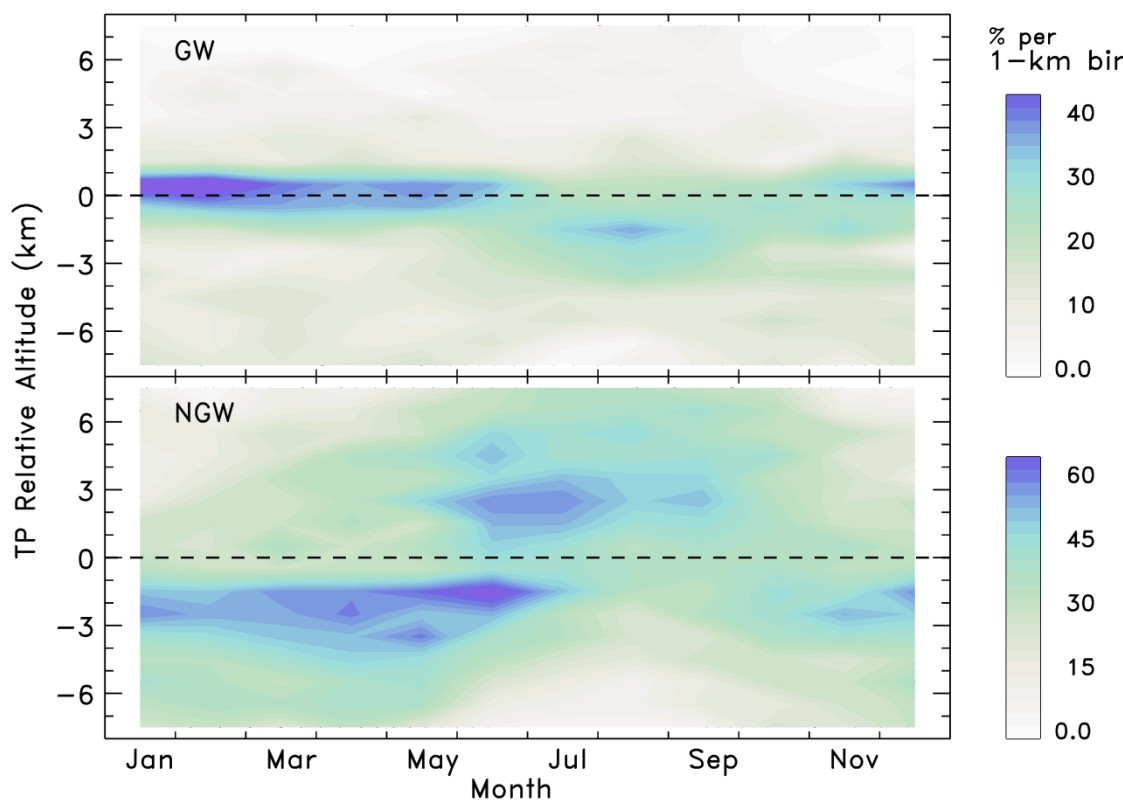

**Figure 9: Climatology of ozone laminae frequency at Boulder, CO, as a function of month and altitude relative to the WMO tropopause for GW (top) and NGW (bottom) laminae. Frequency is expressed as the percent of soundings that contain one or more GW or NGW laminae within 1-km altitude bins relative to the tropopause. Note the different scales between GW and NGW frequencies.**