# Peer review of "Detection and Classification of Laminae in Balloon-borne Ozonesonde Profiles: Application to the Long Term Record from Boulder, Colorado"

_Atmospheric Chemistry and Physics, 2018_

## Referee Comment (RC1) · Anonymous Referee #2 · 31 Oct 2018

General comments:

This manuscript proposed a new algorithm to detect an ozone lamina in the upper troposphere and lower stratosphere (UTLS). Although many previous studies applied different detection methods to ozone vertical profiles in UTLS, most of them suffered from false detection of ozone lamina around the tropopause. The method proposed in this manuscript significantly reduces the false detection of ozone lamina around the tropopause and enables the study of ozone lamina in the whole UTLS region. This method was applied to the massive ozonesonde data at Boulder and provided

seasonal and height variations of ozone lamina and characteristics of its generation mechanisms. In addition, this method can be applied to the gravity wave study near the tropopause, so that it has wide applications. Thus this method has the possibility to provide significant benefit for the atmospheric community. The manuscript is well written and well organized. The topic dealt in this manuscript is suitable for ACP. Description of the method and its evaluation based on the application to the ozonesonde data look convincing in most parts. Thus I recommend a publication of this manuscript after addressing a few specific comments given below.

Specific comments:

The authors tested the application of the method to tropical and midlatitude ozonesonde data. I would like to mention two points to be considered when applying this method to the Antarctic.

- In austral spring, ozone concentration inside the Antarctic ozone hole becomes nearly zero, so that the ozone lamina definition based on a relative amplitude of ozone perturbation will cause false detection of many small-amplitude laminae. Inside the ozone hole, ozone-enhanced layers have been studied as a measure of cross-vortex mixing (e.g., Moustaoui et al., 2003; Tomikawa and Sato, 2010). In order to apply this method to the Antarctic, the ozone lamina definition based on the absolute amplitude of ozone perturbation could be required.

- Thermal tropopause cannot be definitely defined over the Antarctic in winter, because temperature decreases with altitude even in the stratosphere. In this case, it may not be appropriate to use the thermal tropopause as a tropopause definition. On the other hand, ozone tropopause can be clearly defined even in the Antarctic winter (Tomikawa et al., 2009), so that its usage for the tropopause definition could be better.

The authors reported that the GW lamina maximized around the tropopause. Is there a possibility that it was caused by false detection of ozone and potential temperature lamina around the tropopause? It should be discussed in the manuscript.

Is there a plan to disclose "RIO SOL" to the research community? If yes, please mention it in the text.

p.9, l.4

Please put "-" between January and February.

p.11, l.15

"Krizan et al (2016)" should be replaced by "Krizan et al. (2015)".

p.13, l.11

Please put "2011" at the end of this reference.

p.13-16

Isotta et al. (2008), Schmidt et al. (2008), and Thompson et al. (2010) are not cited in the text.

p.17, l.3

"fight" should be replaced by "right".

References:

Moustaoui, M., H. Teitelbaum, and F. P. J. Valero, Ozone laminae inside the Antarctic vortex produced by poleward filaments, Quart. J. Roy. Meteor. Soc., 129, 3121–3136, 2003.

Tomikawa, Y., Y. Nishimura, and T. Yamanouchi, Characteristics of tropopause and tropopause inversion layer in the polar region, SOLA, 5, 141–144, 2009.

Tomikawa, Y., and K. Sato, Ozone enhanced layers in the 2003 Antarctic ozone hole, J. Meteorol. Soc. Japan, 88, 1–14, 2010.

---

## Referee Comment (RC2) · Anonymous Referee #3 · 9 Nov 2018

Review submitted: 9 November 2018

General Comments:

This paper outlines a new approach to the identification of laminae in ozonesonde profiles using the long-term record of profiles from Boulder, Colorado. The authors note the sensitivity of their approach in the UT/LS region is greater than prior approaches. Much of the improvement results from the switch from using an absolute standard for partial pressure/concentration differences in ozone as a function of altitude to a percentage

change – the authors use a 10% criterion. Their result, therefore, is perhaps unsurprising – concentrations in the upper portions of troposphere below the tropopause tend to be lower, perhaps making meeting the absolute standard more difficult.

It would be useful for the authors to show a comparison of the prior approach(es) as applied to the Boulder data set to the current approach to better and more clearly identify the laminae. In the present version, the authors identify gravity wave (GW) and non-gravity wave (NGW) laminae. While the authors note the similarity in the frequency of NGW laminae to Rossby wave frequencies identified in prior papers, it would be useful to use those prior techniques to specify the Rossby wave frequency.

The new technique is applied to a single data set – Boulder. That midlatitude site is located fairly close to and just downwind of the Rocky Mountains. How well does the technique work in other locations? What constraints (if any) are there in application of this new approach? How would this technique do with laminae due to "notches" appearing due to SO2 interference (e.g., Morris et al., 2010, J. Atmos. Ocean. Tech.) in the cathode cell reactions of the ozonesonde?

The authors also note that negative laminae occurred more frequently than positive ones. What is the physical mechanism responsible for this difference in frequency? How does the detection of laminae relate to instrument response time and ascent rate? Given that the magnitudes of the laminae are critical to their detection, if ascent rates are too fast or response times too slow, how will that impact the detection of laminae?

Recommendation: The authors take a novel approach to identification of laminae in ozone profiles. The approach may well compliment past approaches that have been applied in prior papers. As you can see above (and in some detailed comments below), I have some additional detailed questions that the authors should answer before publication. That said, this paper will make a valuable contribution to the literature and should be published.

Specific Comments:

p. 2, line 11 – maybe add the L. Pan et al. (GRL, 2014) paper on stratospheric intrusions associated with convective events.

p. 3, line 4 – data are indeed gathered throughout the entire ∼3-hours of the flight (both ascent and descent), "…at 1-sec resolution during the flight with burst altitudes typically at or above 30 km."

p. 3, line 8 - 9 – "…descent phase of the sounding, descending data are rarely examined; mixing ratio profiles used here…"

p. 4, line 32 – I think Sparling did some work in scale-invariance as well… (e.g., Sparling et al., 2006, JGR).

p. 5, line 9 – "…false laminae…" It would be good to better define how you know they are false detections.

---

## Author Response (AR1)

**Response to Reviews for ACP -2018-884, "Detection and Classification of Laminae in Balloon-borne Ozonesonde Profiles: Application to the Long Term Record from Boulder, Colorado", by Minschwaner et al.**

We thank both reviewers for their detailed assessments and useful suggestions that have led to improvements in the manuscript. Below, our responses (indicated in blue text) follow each of the reviewers' comments.

**Anonymous Referee #2**

The authors tested the application of the method to tropical and midlatitude ozonesonde data. I would
like to mention two points to be considered when applying this method to the Antarctic.

- In austral spring, ozone concentration inside the Antarctic ozone hole becomes nearly zero, so that the ozone lamina definition based on a relative amplitude of ozone perturbation will cause false detection of many small-amplitude laminae. Inside the ozone hole, ozone-enhanced layers have been studied as a measure of cross-vortex mixing (e.g., Moustaoui et al., 2003; Tomikawa and Sato, 2010). In order to
apply this method to the Antarctic, the ozone lamina definition based on the absolute amplitude of ozone perturbation could be required.

- Thermal tropopause cannot be definitely defined over the Antarctic in winter, because temperature decreases with altitude even in the stratosphere. In this case, it may not be appropriate to use the thermal tropopause as a tropopause definition. On the other hand, ozone tropopause can be clearly defined even
in the Antarctic winter (Tomikawa et al., 2009), so that its usage for the tropopause definition could be better.

Reviewer #2 has raised some important issues that should be considered when applying our method to winter-spring soundings in and around the southern polar vortex. We would most likely have to revise our criteria and thresholds for both ozone laminae and the tropopause in order to maintain a robust
analysis under conditions of significant ozone depletion and/or changes to the thermal structure of the lower stratosphere. Note that these issues may also arise in the northern polar vortex during cold winters. On the other hand, a preliminary analyses of other midlatitude and tropical ozonesonde datasets using the standard set of RIO SOL parameters has been done, and the results appear to be similar in accuracy to those from Boulder. A detailed comparison of ozone laminae from different
measurement sites is planned for future investigations, but in this paper, our emphasis is on a description of techniques and a general climatology from Boulder. As both reviewers have noted the obvious extension to datasets from other sites (see also response to Reviewer #3 below), the manuscript has been revised to include a discussion of the applicability of using RIO SOL with the current set of parameters for analysis of soundings from other locations (p. 9).

The authors reported that the GW lamina maximized around the tropopause. Is there a possibility that it was caused by false detection of ozone and potential temperature lamina around the tropopause? It should be discussed in the manuscript.

We examined this possibility and have concluded that the use of a narrower width filter in the vicinity of the tropopause effectively eliminates false detections due to sharp gradient changes, under most tropopause conditions. On p.6 we stated that there was no altitude dependence in the frequency of false detections in our simulated lamina tests. New text has been added on p.8 to stress this point in connection with our discussion of the maximum in laminae frequency seen just above the tropopause.

Is there a plan to disclose "RIO SOL" to the research community? If yes, please mention it in the text.

RIO SOL and its documentation are available upon request from the lead author. A short section on "Data and Code Availability" has been added after the Conclusion section.

p.9, l.4 Please put "-" between January and February.

p.11, l.15 "Krizan et al (2016)" should be replaced by "Krizan et al. (2015)".

p.13, l.11 Please put "2011" at the end of this reference.

p.13-16 Isotta et al. (2008), Schmidt et al. (2008), and Thompson et al. (2010) are not cited in the text.

p.17, l.3 "fight" should be replaced by "right".

All of the above changes have been implemented, Isotta et al. (2008) is now cited in the text, and the Thompson et al (2010) reference has been deleted. Note Schmidt et al. (2008) was already cited, but Schmidt et al. (2006) was not cited and that reference has also been deleted.

**Anonymous Referee #3**

It would be useful for the authors to show a comparison of the prior approach(es) as applied to the Boulder data set to the current approach to better and more clearly identify the laminae. In the present version, the authors identify gravity wave (GW) and non-gravity wave (NGW) laminae. While the authors note the similarity in the frequency of NGW laminae to Rossby wave frequencies identified in
prior papers, it would be useful to use those prior techniques to specify the Rossby wave frequency.

A comparison between RIO SOL and a prior approach for a single ozonesonde profile is presented in Figure 2, and we have also added a new paragraph discussing the Boulder laminae climatology from RIO SOL in comparison to a fixed-width filter and correlation window technique (p. 9). In terms of classifying "Rossby wave laminae", we fail to see the physical mechanism(s) underlying their
identification using correlation thresholds between ozone and potential temperature, as has been done in prior papers. While we expect some fraction of NGW ozone laminae are, in fact, associated with Rossby wave activity, their connection (or lack thereof) with perturbations in $\Theta$ is not as clear as in the case of GW laminae. We have attempted to clarify this view with new text on p. 11.

The new technique is applied to a single data set – Boulder. That midlatitude site is located fairly close
to and just downwind of the Rocky Mountains. How well does the technique work in other locations? What constraints (if any) are there in application of this new approach?

Both of these issues - constraints on the application of RIO SOL, and suitability for other locations, are now addressed on p. 9 of the manuscript (see also response to Reviewer 2 above).

How would this technique do with laminae due to "notches" appearing due to SO2 interference (e.g.,
Morris et al., 2010, J. Atmos. Ocean. Tech.) in the cathode cell reactions of the ozonesonde?

Any process which leads to a perturbation in the ozonesonde profile, whether real or an artifact in the measurement, will also lead to lamina identification in RIO SOL if the amplitude exceeds 10% and the vertical scale is between 0.2 and ~2 km. This is now discussed on p. 7.

The authors also note that negative laminae occurred more frequently than positive ones. What is the physical mechanism responsible for this difference in frequency?

The physical mechanism responsible for the higher frequency of negative laminae is not definitively known. Perhaps in future studies, when processes leading to the generation of NGW laminae are more clearly identified, the reasons for this difference will become clearer.

How does the detection of laminae relate to instrument response time and ascent rate? Given that the magnitudes of the laminae are critical to their detection, if ascent rates are too fast or response times too slow, how will that impact the detection of laminae?

The detection of laminae is not significantly impacted by ascent rates and response time under most conditions. However, we do anticipate a possible systematic offset in laminae altitudes on the order of 100 m. A discussion of this effect has been added on p. 7.

**Track changes, revised manuscript follows**

[revised manuscript text omitted]

One factor that may introduce a systematic offset to laminae central altitudes is the finite response time of the ozonesonde. Sonde ascent rates are consistently 4-6 m/s and response timescales are ~25 s, leading to a possible systematic bias of between +100 m to +150 m in altitude. Note that the offset is unlikely to be any bigger than this because we consistently find gravity wave laminae for which the ozone and $\Theta$ perturbations (which are based on temperature with response times on the order of a few seconds) are very well correlated on a 100-m grid, with no systematic altitude offsets. Given the 1-Hz sampling for the raw data, the effect of variations in ascent rate which can act to smooth measured profiles or to limit the resolving of laminar features, with scales greater than 0.2 km, is minimal. There are also rare, but documented (e.g. Morris et al., 2010), measurement artifacts that could be mistakenly identified as ozone laminae. In the case of $SO_2$ interference observed by Morris et al., RIO SOL would interpret apparent ozone "notches" as negative NGW ozone laminae.

**4 Results**

**4.1 Overall Statistics for Ozone Laminae**

[revised manuscript text omitted]

The above statistics for Boulder may be contrasted with those obtained using previous methodologies, namely the use of a fixed width, 6-km boxcar filter for the basic state and a 5-km wide correlation window for $O_3$ and $\Theta$ anomalies. For this particular method, the total number of laminae detected is 25% smaller and their mean widths are more than twice as large as those from RIO SOL shown in Figure 5, ranging from 1km up to a maximum of 3 km at the tropopause. These differences can be attributed to the dominating influence of tropopause-induced laminae for the case of a fixed width boxcar. The apparent tropopause lamina appears in over half of the soundings and it is sufficiently wide to mask or absorb any other individual laminae that may be present within 2-3 km of the tropopause. Not surprisingly, we also detect fewer overall positive laminae when using the fixed-width boxcar. There are also changes to the relative fraction of GW and NGW laminae when using a 5-km wide correlation window; relatively more GW laminae are detected and this fraction maximizes at the tropopause level in association with the aforementioned spurious tropopause-induced laminae in both $O_3$ and $\Theta$.

A preliminary analysis has also been done using RIO SOL in its standard configuration for other measurement sites at mid and low latitude stations, and the results appear to be similar in accuracy to those from Boulder. At Pago Pago, Samoa, we find comparable statistics with a slightly higher overall frequency of laminae (nearly 10 per profile), a 30% GW fraction, and a similar altitude distribution relative to the tropopause. On the other hand, when applying RIO SOL to winter-spring soundings in and around the Antarctic polar vortex or during particular cold periods in the Arctic winter, the criteria and thresholds for both ozone laminae and the tropopause would likely require significant changes in order to maintain a robust analysis, especially under conditions of significant ozone depletion and/or changes to the thermal structure of the lower stratosphere. A detailed comparison of ozone laminae from different measurement sites is planned for future investigations, but in this paper, our emphasis is on a description of techniques and on the climatology from Boulder.

**4.2 Basic States and Ozone Variance**

[revised manuscript text omitted]

The distribution of NGW laminae stands in stark contrast to that of GW laminae, and, as noted in the introduction, the generating mechanisms for NGW laminae are much more uncertain. Previous analyses have used maximum correlation threshold criteria between ozone and potential temperature to infer the influence of Rossby waves on ozone (e.g. Pierce and Grant 1998; Thompson et al 2007a). We have not adopted this approach in RIO SOL due to larger uncertainties in positively classifying these laminae using ozone and potential temperature alone, particularly since the connection between $O_3$ and $\Theta$ is not as clear as indicated in eq. (1) for GW ozone laminae. However, While we have not adopted this approach in RIO SOL, it is plausible that a significant fraction of NGW laminae are generated in processes associated with Rossby wave activity. The seasonality of Rossby wave breaking in the UTLS at northern midlatitudes (e.g., Hitchman and Huesmann, 2007; Isotta et al., 2008) is similar to the seasonality of UT NGW ozone laminae seen in Figure 9. Rossby wave breaking and stratosphere/troposphere exchange (STE) are both prevalent along the flanks of UT jets (Gettelman et al., 2011 and references therein), and studies of jets using meteorological reanalysis data show a maximum in subtropical UT jet frequencies near 30$^o$ latitude from November through April in the Northern Hemisphere (Manney et al., 2011; 2014). An associated phenomena of multiple tropopause events, which are often linked to tropopause folds and extratropical STE (e.g., Sprenger et al., 2003), also show maximum frequencies during December-March on the northern flank of the subtropical jet maximum (e.g., Manney et al., 2014), and these events are responsible for a significant fraction of the variability in ozone and other trace gases in the tropopause region (Schwartz et al., 2015).

Mechanisms leading to the maximum in NGW laminae frequency in the LS during summer are more uncertain. Jing and Banerjee (2018) found a maximum in anticyclonic RWB on the 350 K and 370 K $\Theta$ surfaces during NH summer, and both of these surfaces are typically at or above the tropopause level over Boulder during summer. In addition, they showed that the zonal distribution of summer RWB favored those regions above and immediately upstream of Boulder. However, the production of laminar featuress in summertime LS ozone from other mechanisms cannot be ignored. Impacts on the composition of the midlatitude summer LS have been demonstrated from monsoon-related dynamics (e.g., Randel et al., 2010), deep summertime convection (e.g., Weinstock et al., 2007), and meridional transport from the tropical UT (e.g., Bönisch et al., 2009). Clearly, more work is needed to understand the full range of dynamical and chemical processes that may produce the range of laminar ozone features seen in these balloon profiles and in other ozone measurements. The development of methods to further classify NGW laminae and to associate mechanisms with their generation is a focus of ongoing work using RIO SOL.

**5 Conclusions**

We have described the RIO SOL analysis package for characterizing ozone laminae in balloon soundings and presented an analysis of the ~25 year ozonesonde dataset from Boulder, Colorado. RIO SOL involves an adaptation of the filter and difference approach used in previous studies of ozonesonde profiles, in which a unique basic state is generated for each ozone profile and laminae are identified as deviations from this basic state. The major improvements in RIO SOL include methods for improved sensitivity in identifying GW laminae using potential temperature from each sounding, and for avoiding false detections of GW laminae near the thermal tropopause. The vertical gridding of the ozonesonde data and the filtering method constrain the range of vertical scales for identified laminae to between 0.2 and 2.5 km. Simulations indicate that RIO SOL can reliably identify most of the ozone laminae with relative amplitudes greater than 10%, and virtually all laminae above 20% amplitude.

[revised manuscript text omitted]